# Development of Dynamic Capabilities for Automotive Industry Performance under Disruptive Innovation

**Atichat Rotjanakorn [1],\*, Pornrat Sadangharn [1],\* and Khahan Na-Nan [2],\*** 

[1]   Faculty of Management and Tourism, Burapha University, Chonburi 20131, Thailand
[2]   Faculty of Business Administration, Rajamangala University of Technology Thanyaburi,
      Pathum Thani 12110, Thailand
\*    Correspondence: atichat.rotjanakorn@gmail.com (A.R.); pornrat@buu.ac.th (P.S.);
      khahan_n@rmutt.ac.th (K.N.-N.)

**Abstract:** Dynamic capabilities are creating dramatic change for the industry around the world. Resource-Based View (RBV) theory and Operational capability theory are the basic capabilities of an organization under a normal changing environment. This creates a competitive advantage and organizational success in a relatively short period of time, in which the dynamic environment is not sufficient to cope with this change. Dynamic capability is a concept for managing change under this dynamic environment. Past research supports a direct positive relationship between dynamic capability and firm performance but it did not focus on the mediator variables. This research emphasizes the influences of competitive advantages and innovation capabilities as mediators of dynamic capabilities and firm performance were investigated. A cross-sectional design study was utilised and questionnaires were submitted to 326 firms to test the proposed relationships. IBM SPSS Statistics Base 26, IBM SPSS AMOS 21, and PROCESS macro 3.6 were used for statistical analysis. Results revealed that competitive advantages and innovation capabilities were partially mediated by dynamic capabilities and firm performance. Findings contribute to the literature on empowering leadership and innovative firm performance by highlighting that competitive advantages and innovation capabilities act as mediators to improve dynamic capabilities and enhance innovative firm performance.

**Keywords:** dynamic capabilities; competitive advantages; innovation capabilities; firm performance

## 1. Introduction

### 1.1. Reason of Research

The automotive industry is a major industry that has driven the economy of Thailand, which has been supported by the government since the Second National Economic and Social Development Plan. Thailand has become one of the world's major automotive carmakers and automotive parts manufacturers. Due to the rapidly changing air pollution and natural environment, people worldwide are looking for vehicles that are more energy-efficient and environmentally friendly. For this reason, the demand for vehicles in the world market has changed from the internal combustion engine vehicles, in which Thailand has expertise in both assembly and production of the change, to the new demand for electric vehicles in Thailand, which is still in the period of development plan only. Electric vehicles are seen as a disruptive technology that will make a difference and affect the automotive industry worldwide. The supply chain of the Thai automotive industry has to face the modern technology that comes with electric vehicles. The Resource-Based View (RBV) and Operation Capability that the

organization has under normal conditions may not be sufficient under the environmental turbulence situation. Dynamic capability is therefore an important role in managing the organization's strategy to be suitable for competitive and highly dynamic situations. This article aims to present guidelines for the development of dynamic capability in the Thai automotive industry, in terms of sensing, seizing, and transforming to be a guideline for preparing the adaptation of the Thai automotive industry to keep up with changes that will occur in the future.

Recent rapid changes in the social and economic aspects of competition in the industry have resulted in the resource-based view (RBV) and operation capability theories as basic management principles of organisations under normal conditions. However, RBV is not sufficient as an efficient tool for executive decision-making and problem-solving in every situation. This research emphasises the dynamic capabilities that affect performance. Results show the importance of dynamic capabilities on firm performance by passing on innovation capabilities and competitive advantages as partial mediators. Thus, high dynamic capability in terms of the ability to sense, seize, and transform new ideas will lead to prompt and effective changes in company strategies [1]. By creating competitive advantages over competitors in terms of cost leadership and differentiation or applying innovations in design and manufacturing, firm performance can be optimised under changing environments. Therefore, for a company to survive, it must adapt by creating competitive advantages and adopt innovations for both products and services.

Resources and capabilities are factors that enable companies to compete and lead to competitive advantages. Winter [2] defined zero-level capabilities as the ability of a company to "earn a living" as a static view that did not create any competitive advantages. The ability to change products and production processes and respond to changing customer needs can be called the dynamic capabilities of a company and requires integration of learning and workstyle adjustment [3]. Thus, dynamic capabilities are more powerful than basic attributes. Wilden and Gudergan [4] showed that resource-based views and operational capabilities have a major influence on firm performance. The theory of resource-based views and operational capabilities concerns the ability to quickly create competitive advantages to keep pace with rapid changes [5–7]. This research addresses the dynamic capabilities that are important when creating competitive advantages as innovation capabilities that will lead to success of the Thai automotive industry. Findings will benefit the Thai automotive industry as a guideline for adaptation and future development.

### 1.2. Research Questions

Firm performance studies are important and urgently required to expand the body of knowledge. However, currently available information to better understand the different phenomena and enhance firm performance is inadequate. Accordingly, study variables were examined to confirm their validity and reliability to explain firm performance. Two main research questions were postulated: (a) Are competitive advantages (CPT) and innovation capabilities (INC) mediators between dynamic capabilities (DYC) and firm performance (FRM)? and (b) How does DYC influence FRM? Results will be useful for researchers, educators, students, human resource officers and the general public who are interested in studying problems with performance appraisal.

## 2. Theoretical Background

### 2.1. Dynamic Capabilities

Dynamic capabilities (DYC) have unique characteristics that are considered an important part of any company. Strong dynamic capabilities can build the capacity necessary to deal with the growing uncertainty of innovation and competition in the current market. Teece and Leih [8] and Winter [2] noted that dynamic capabilities differ from a resource-based view (RBV). Dynamic capabilities place great importance on environmental changes, but a RBV focuses on the management of resources that are available under normal conditions. General competency relates to the skills required to perform duties

that are appropriate for management and corporate governance; however, dynamic capabilities create and adapt to a business environment that is better suited to change. Protogerou and Caloghirou [9] conducted a quantitative study of 271 organisations from various sectors. They concluded that dynamic capabilities involved coordination of learning as a positive response to competition and had a great impact on the ability of a company to operate in a changing technological and marketing environment.

Dynamic capabilities can be explained as the sum of the definitions of each word. Dynamic is the result of creating new things in line with the changing environment, while capabilities involve integrating new skills and knowledge that regularly and continuously provide work expertise and reconfigure to the changing external environment [10–12]. Dynamic capabilities refer to the ability of an organisation to reduce connections and use available resources to create new things in accordance with the changing environment. Dynamic capabilities can be used to summarise and create conceptual variables [3,10,13] with the following components:

(1) Sensing (SEN) is the ability to first realise and then learn and understand information concerning changes occurring through the fluctuating and unpredictable risk of technology. Responding to these situations can conceptually solve problems. Sensing can be likened to the facility to foresee the future and adapt by developing new abilities [14].

(2) Seizing (SEI) is the ability to choose resources to match the changes in opportunities. Strategic operation plans change and adapt through learning skills to match new customer requirements and use technology to recognise changes that will occur. Cost adjustments are made to adjust the product to meet different future needs of consumers [14].

(3) Transforming (TRN) is the ability to adjust, reduce, or increase company resources in accordance with continuous change. Existing knowledge is applied and combined with new knowledge to develop products and processes systematically. New knowledge is transferred to a format that employees can easily understand [14].

Dynamic capability is a key building organizational innovation capacity to promote creativity and performance. Organization that focuses on the development of strong dynamic capability will be able to drive strategy and adapt to innovation, making it able to compete with competitors [15–17]. Zhou and Zhou [18] studied the relationship between dynamic capability and organizational performance. By collecting data from 204 Chinese companies, the study found that innovation capability is a mediator between dynamic capability and organizational performance. In addition, Zahra's [19] research shows that organizations competing in a market with high volatility and growth pay more attention to innovation capability by offering the new products and processes, compared to organizations that compete in a market that is not growing, where there is not much competition in the industry.

## 2.2. Firm Performance

Firm performance (FRM) theory refers to a balanced evaluation (balanced scorecard) as a management tool that helps to manage and implement strategies based on the concept of balanced organisational evaluation. Kaplan and Norton [20] proposed a scenario of organisational evaluation that considered other factors apart from financial indicators, as follows:

(1) Financial perspective is an indicator of how strategies are implemented and how to increase profitability and attain the desired goals. Survival is measured by cash flow, success is measured by sales growth and net profit, while progress is made by increasing investments in other businesses.

(2) Customer perspective is the ability to respond to customer needs. Goals must be clearly set in terms of time, quality, usability, and service. The key objectives are to retain existing customers, add new customers, maintain customer satisfaction, and increase market share.

(3) Internal processes as company working systems are important from both financial and customer perspectives. Competent internal development and management results in an efficient production process with high-quality products distributed on time to match customers' needs following standards that comply with regulations.

(4)   Learning and development perspectives comprise the ability to develop new knowledge that is directly related to the value of products delivered to customers. Information systems are continuously improved with updated technology and internal readiness to accommodate changes that may occur.

DYC and FRM are related. Increasing evidence suggests that DYC has a big impact on FRM. Henderson and Cockburn [21] found that organisational ability to gather knowledge from external sources had a positive relationship with productivity and performance. Singh and Zollo [22] studied the integration process after asset acquisition in the banking sector. They provided evidence that buyers who invest more effort in an organisation's integration process are likely to achieve superior profits compared to competitors. The organization is able to maintain its financial flexibility [23]. The volatile business environment also forces the organization to use its internal resources to increase the capacity to introduce new products and services. Empirical studies have shown that strategies have a positive influence on process innovation and product innovation [24–26]; innovation strategies such as the vision of organizational innovation, cooperation in finding results with customer needs in mind by an innovative strategy-based approach. Many of the studies have attempted to examine the relationship between innovation capability and performance. The analysed studies yield different results, although a positive link between innovation capability and performance is generally confirmed [27], but process innovation can lead to organizational performance by reducing costs and increasing production volumes, which reduces product development time.

Collis [28] suggested that DYC-assisted organisations avoid using outdated and non-standard guidelines. Thus, organisations must develop the ability to learn and define new resource bases to overcome the traps set by existing capabilities and create a new source of competitive advantages. Eisenhardt and Martin [29] drew the same conclusion using different arguments. Although DYC may be considered valuable and rare, they are important characteristics and cannot be imitated. DYC cannot maintain sustainable competitive advantages; they will contribute to success by consolidating and renewing the ability to affect FRM. Overall, results confirmed that DYC created and configured a new resource location.

The relationship between dynamic capability and innovation capability can stimulate an increase in organizational innovation activities, leading to successful performance [30,31].

The dynamic customer demand environment allows organizations to strive to deliver ideas and innovative products or services that can create greater value [32]. An organization's innovation strategy is based on the capacity and knowledge the organization has at the time. The dynamic capability and innovation capabilities of an organization are strongly related to innovation strategies, and both are based on the resources generated internally and externally [33].

Therefore, if an organisation wishes to increase dynamic capabilities, the external environment must first be considered to create a new process that can be used regularly by managing existing resources. This demonstrates the integration, creation, and structural adjustment required to assess risks and market opportunities [34]. In summary, dynamic capabilities are high-level activities that help organisations set goals for the production of existing products and services that have high market demand [8,35]. Based on the literature review, the first research hypothesis was postulated as below.

**Hypothesis 1 (H1).** *DYC directly affects FRM.*

*2.3. Competitive Advantages*

Competitive advantages (CPT) refer to superiority over other organisations by creating different capabilities over competitors in the industry [36]. Strategies must be adopted that can change and adapt to resources and business readiness [37]. Two strategies of creating competitive advantages are as follows:

(1) Cost leadership manages resources through cost control to be less than competitors with least loss. While still providing equal benefits to resources that competitors use, most companies focus on ways to reduce costs rather than increasing sales prices, leading to lower product prices than competitors.

(2) Creating product value and differentiating from competitors is a strategy that promotes resource investment to adjust the product to meet the needs of customers.

DYC are important to maintain competitive advantages [38,39]. Future competition and the industrial environment are difficult to predict. Organisations need to be flexible, especially with regard to the appropriate time for entering the market and making informed decision changes. In response to today's ever-changing environment [40], O'Reilly III and Tushman [41] suggested that DYC can consolidate organisational resources at lower costs with higher asset utilisation. This increases CPT in response to environmental changes in the industry. These findings indicate that DYC has a significant positive influence on CPT, which can be defined as implementing strategies to reduce costs and differentiate products or services. Using opportunities that change with the environment increases general efficiency and company performance in accordance with targeted strategy [42,43].

Aguirre [44] studied the DYC and CPT of Mexican companies and concluded that DYC and CPT were important for survival in an innovative and rapidly changing technological environment. Organisations should stimulate DYC to become competitive. DYC and CPT cannot be viewed separately as new talents emerge in a changing environment [45]. Anwar [42] studied the impact of competitive advantage on the performance of Small and Medium Enterprise (SMEs) from the manufacturing sector in the Pakistani market. Data were collected from 303 companies. The results showed that competitive advantage had a significant influence on the performance of SMEs. Competitive advantage is defined as implementing a cost reduction strategy and differentiating products or services. It takes advantage of the opportunities that change according to the environment to increase the organization's efficiency, affecting the organization's target [43]. Dynamic competence was positively correlated with organizational competitive advantage and performance [46]. Udriyah and Tham [47], Songling and Ishtiaq [48], and Cantele and Zardini [49] studied the competitive advantage as a variable transmission on the performance of the organization. The research found that competitive advantage has a significant influence on the organizational performance. Navarro-García and Moreno [50] studied the relationship between dynamic capacity and organizational performance with competitive advantage as a transmission factor in the car sales business. The results of the research showed that dynamic capacity influenced the organization's performance through a variable of competitive advantage.

Future competition and the overall industrial environment are unpredictable; therefore, the timing of market entry and decision-making must be flexible to keep pace with changes in response to today's ever-changing environment [40]. Dynamic companies can gather and preserve their resources, resulting in reduced costs and high-efficiency asset utilisation. Therefore, dynamic capabilities have a significantly positive influence on competitive advantages [41,51–53]. Based on the literature review, the second research hypothesis was postulated as below.

**Hypothesis 2 (H2).** *CPT acts as a mediator that transmits the effects of DYC to FRM.*

*2.4. Innovation Capabilities*

The concept of innovation capabilities (INC) refers to creative thinking. This leads to strong organisational structure, culture, and human resource management to create new products, services, and new work methods [54].

(1) Product innovation presents new items and improves the efficiency of existing material to truly meet the needs and reach target customers faster than competitors.

(2)　Process innovation generates a new work process or improves an existing process to respond to strategic goals in a timely manner faster than competitors.

Current scholars focus on innovation as a process that is continuously repeatable and can be displayed in a variety of formats. The aim of the innovation is to use the latest conditions and opportunities that have been formed in the environment to redefine value and gain a competitive advantage [55,56]. Its permanent use and innovation provide a highly competitive advantage for the entire organization. So, innovation in today's global market is fundamental to inevitable change [57]. Therefore, innovation is generally a process that is seen as a goal of innovation in different activities, by creating new products, service creation, brand new, installation of new management systems, new economic or public value creation, etc.

Innovation capabilities is a variable that transmits the relationship between dynamic ability and organisational performance [18]. Dynamic customers demand change and improved organisational performance. Both strive for new ideas and innovative products or services that can create value for the organisation [32]. Company innovation strategy depends on available knowledge and workforce ability. Both dynamic capabilities and innovation capabilities are, therefore, highly correlated with innovation strategy to deliver successful and effective results [30,31,33]. Companies that have grown and are able to compete in a highly volatile environment focus on strategies of innovation capabilities [19] that have a positive influence on process and product innovation [24,26]. Dynamic innovation has two situations. First, in a highly innovative situation that usually happens at the early stage of industry, the organization will accelerate the development of innovations to keep pace with the advent of technology and customers' needs by focusing on cost reduction and selling prices; second, in less dynamic innovative situations of a matured market. Organization of all competing firms does not show a significant innovation difference but pays more attention to the price competition [58], and the industry continuously adopts open platforms to create and maintain ecosystem innovation [59]. The production was positively correlated with short-term overall productivity increase and long-term total yield growth under dynamic innovation situations. In addition, promoting technology and innovation management and supporting research and development subsidies may reduce incremental costs of conducting research and development and increase technology and innovation management rates and R&D activities [60,61].

Dynamic competence is an important factor in building innovation capacity by promoting creativity to achieve company goals. Organisations that focus on strengthening dynamic development will be able to drive strategies and adapt to innovation, allowing them to compete with competitors quickly [15,16]. There is generally a positive link between innovation capabilities and organisational performance [27], but process innovation can also create good results. Product innovation helps companies to continuously improve, resulting in more efficient financial performance [62]. In the dynamic innovation situation, the economic growth rate will be faster or slower depending on balance innovation capability among the three sub-economies such as SME, start-up, big business, and social open innovation [63]. DYC should be applied in sub-economies in balancing innovation capability to all cycling. Zhou, Zhou [18] suggested that innovation serves as an important mediating mechanism between dynamic capabilities and performance. Giniuniene and Jurksiene's [64] findings on innovation should be analyzed as a mediator in the relationship between competence and performance; organizational learning is also a necessary variable.

The objective of this article is to resolve this controversy by proposing and testing a mediating mechanism between the dynamic capacity and empirical performance of a company. The extant literature suggests that dynamic capabilities adaptation not enough for the pattern of environmental change. This is because innovation is considered to be a key mechanism for companies to adapt and shape the environment in which they operate [29]. Based on the literature review, the first and second research hypotheses were postulated:

**Hypothesis 3 (H3).** *INC acts as a mediator that transmits the effects of DYC to FRM.*

To manage change and maximise capability under environmental fluctuations, the automotive industry should focus on the theory of dynamic capabilities. Previous research showed that the RBV and operational capability theories influenced company performance. However, these theories only provide a single competitive advantage for a short time and cannot keep pace with rapid changes [5–7]. They offer only basic capabilities under normal environmental conditions [7,65], while dynamic capabilities are the new trend in the Thai automotive industry to generate rapid disruptive innovation.

Therefore, this research studied important dynamic capabilities to create competitive advantages and innovation capabilities for the successful future performance of the automotive industry. Findings will be beneficial to the Thai automotive industry as a guideline for adjustment and further future development. Developing and applying knowledge of DYC, CPT, and INC will lead to improved FRM in line with set strategic goals. A conceptual framework showing the association of DYC with CPT, INC, and FRM is shown in Figure 1.

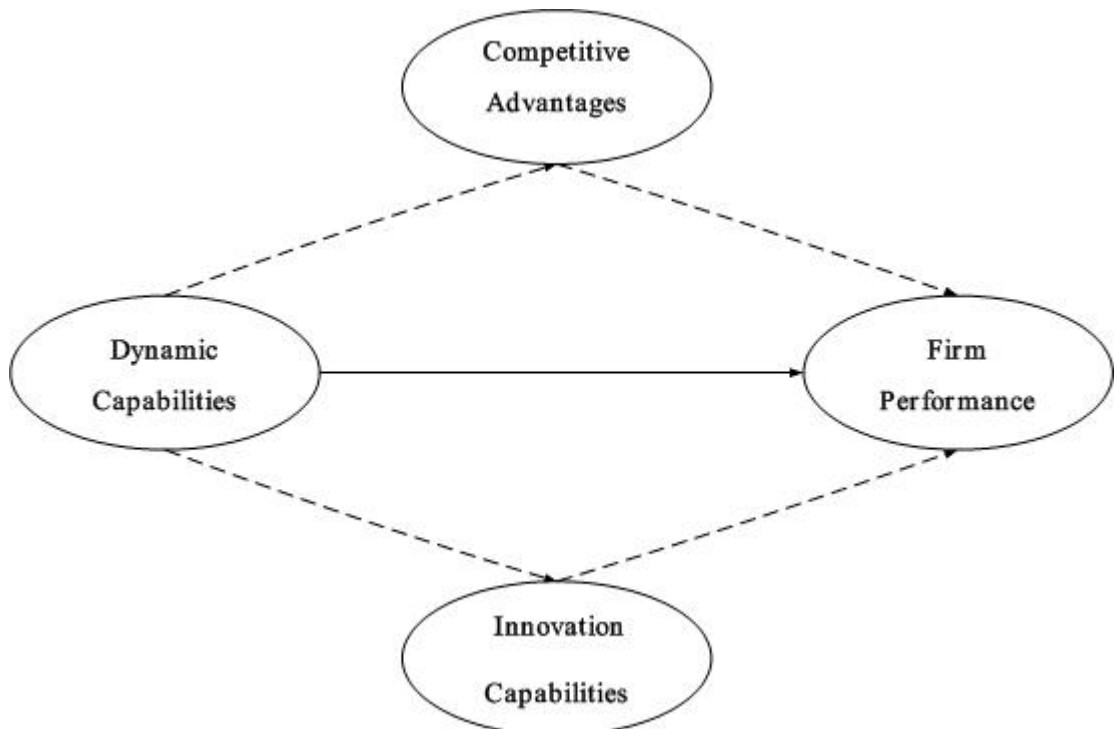

**Figure 1.** Conceptual research framework.

## 3. Research Methodology

### 3.1. Sample and Data Collection

The study population was 833 companies from the Thai automotive industry (data as of 29 July 2019). The sample size of 220 companies was determined using the criteria of Hair, Black [66]. They specified that the ratio between the number of samples and the number of parameters should be 25:1 to be sufficient for confirmatory factor analysis (CFA). Eleven observable variables were identified. Therefore, the sample group must be at least 275 companies. For convenience of evaluation and data analysis, the sample size was finalised at 326 companies. The sample size of 326 companies was selected by use of stratified random sampling. The calculation resulted in 275 sample units. Convenience sampling was followed by email as an effective way for supporting the variety and qualifications of respondents. An online survey has more advantages than post or paper surveys because of the capability for quick replies without the limitation of geography. Therefore, for the first round, 400 survey emails were sent to all samples and 281 replies were received within four weeks.

Survey emails were then sent to another 100 samples, and 45 replies were sent back within four weeks, to give a total of 326 respondents.

A questionnaire with closed-end questions was designed from the study of data from textbooks, documents, concepts, theories, and related research. All indicators were adopted from previous research and measured using a five-point Likert scale (Table 1).

**Table 1.** Questionnaire factors and measurements.

| Factors | Measurements | Previous Research |
|---|---|---|
| Dynamic capabilities (DYC) | Sensing (SEN) Seizing (SEI) Transforming (TRN) | Li, Li [67] |
| Competitive advantages (CPT) | Cost leadership (CST) Differentiate (DIF) | Justinian [68], Susanti, and Arief [69] |
| Innovation capabilities (INC) | Product innovation (PDI) Process innovation (PCI) | Pérez-De-Lema, Hansen [70], Wang, and Ahmed [71] |
| Firm performance (FRM) | Financial perspective (FIP) Customer perspective (CUP) Internal process perspective (INP) Learning perspective (LNP) | Elbanna, Eid [72] |

- Analysis of descriptive statistical data included frequency distribution, percentage, qualifications of the sample, and the Pearson correlation coefficient.
- PROCESS as an extension of SPSS was used to analyse all the assumptions, test the direct influence on firm performance, and also test the indirect effects of both mediator variables.

*3.2. Measurement Validity and Reliability*

All measures for studying factors were fitted from the existing literature to ensure that they were suitable as contextual models, samples, and environments for the conceptual model. The survey was translated into English and Thai as a high-quality translation for investigation by experts in English and organisational behaviour.

To assess the accuracy of the content, five experts on organisational behaviour, management, industry, psychology, and human resource development were recruited to test, rate, and examine all measures. Results indicated that the accuracy of the content varied between 0.8 and 1.00. Rovinelli and Hambleton [73] considered product-intent consistency as item objective congruence (IOC) to be statistically significant for an accuracy test by five experts. The reliability of the questionnaire was less than 0.80, including the survey sample that was validated by considering the internal consistency of the scale with another group of 30 car companies. The correlation coefficient was calculated at 0.962 and all correlations were between 0.465 and 0.831. Streiner and McCave [74] stated that a total correlation value of above 0.20 represented a good list of questions. As a result, each item had a very good correlation. Cronbach's coefficient alpha was used to check the reliability of the instrument and the internal integrity of product measurement and rating scales. Bonett and Wright [75] determined that a correlation coefficient for each study item at higher than 0.946 indicated that the item was internally consistent and the measurement was reliable.

Scale verification was based on the concept of Fornell and Larcker [76]. Confirmatory factor analysis was used to test whether the structural validity of each variable in the model was true, based on empirical evidence of theory and concepts. Statistical consistency was determined by Chi-square ($\chi2$), chi-square/degrees of freedom ($\chi2$/df), goodness-of-fit index (GFI), adjusted goodness-of-fit index (AGFI), comparative fit index (CFI), standardised root-mean-square residual (SRMR) and root-mean-square error of approximation (RMSEA) [77]. Table 2 shows the effectiveness of the composition from the loading values of standard factors for each item or observed variable. All questions

were significantly correlated with structural theory because of the large factor loadings (>0.50) and had significance values at $p < 0.01$ (all $t$ values are 3).

**Table 2.** Confirmatory factor analysis.

| Item | Standardised Factor Loading | $t$-Value | Average Variance Extracted (AVE) and Composite Reliability ($\alpha$) |
|---|---|---|---|
| Sensing (SEN) Model fit indices: $\chi^2 = 0.692$, df = 1, $p = 0.406$, $\chi^2$/df = 0.692, GFI = 0.999, AGFI = 1.0, CFI = 0.004, RMSEA = 0.000 and SRMR = 0.004 | | | |
| SEN1 | 0.86 | 9.28 | $\alpha$ = 0.79 |
| SEN2 | 0.82 | 9.41 | AVE = 0.50 |
| SEN3 | 0.47 | 8.50 | MSV = 0.48 |
| SEN4 | 0.55 | | ASV = 0.35 |
| Seizing (SEI) Model fit indices: $\chi^2 = 10.575$, df = 8, $p = 0.227$, $\chi^2$/df = 1.322, GFI = 0.991, AGFI = 0.976, CFI = 0.996, RMSEA = 0.029 and SRMR = 0.010 | | | |
| SEI1 | 0.59 | 9.84 | $\alpha$ = 0.80 |
| SEI2 | 0.60 | 9.92 | AVE = 0.51 |
| SEI3 | 0.83 | 11.84 | MSV = 0.50 |
| SEI4 | 0.80 | | ASV = 0.35 |
| Model 1: Transforming (TRN) Model fit indices: $\chi^2 = 3.355$, df = 2, $p = 0.187$, $\chi^2$/df = 1.677, GFI = 0.959, AGFI = 0.974, CFI = 0.997, RMSEA = 0.046 and SRMR = 0.009 | | | |
| TRN1 | 0.71 | 10.50 | $\alpha$ = 0.82 |
| TRN2 | 0.80 | 11.30 | AVE = 0.54 |
| TRN3 | 0.76 | 11.02 | MSV = 0.50 |
| TRN4 | 0.66 | | ASV = 0.26 |
| Cost leadership (CST) Model fit indices: $\chi^2 = 1.831$, df =2, $p = 0.4$, $\chi^2$/df = 0.915, GFI = 0.997, AGFI = 0.986, CFI = 1.0, RMSEA = 0.000 and SRMR = 0.013 | | | |
| CST1 | 0.76 | 7.00 | $\alpha$ = 0.79 |
| CST2 | 0.72 | 11.42 | AVE = 0.50 |
| CST3 | 0.85 | 11.74 | MSV = 0.44 |
| CST4 | 0.43 | | ASV = 0.13 |
| Differentiate (DIF) Model fit indices: $\chi^2 = 1.274$, df = 2, $p = 0.529$, $\chi^2$/df = 0.637, GFI = 0.998, AGFI = 0.990, CFI = 1.00, RMSEA = 0.000 and SRMR = 0.009 | | | |
| DIF1 | 0.43 | 7.00 | $\alpha$ = 0.78 |
| DIF2 | 0.74 | 11.42 | AVE = 0.48 |
| DIF3 | 0.81 | 11.74 | MSV = 0.42 |
| DIF4 | 0.73 | | ASV = 0.09 |

**Table 2.** *Cont.*

| Item | Standardised Factor Loading | *t*-Value | Average Variance Extracted (AVE) and Composite Reliability ($\alpha$) |
|---|---|---|---|
| Product innovation (PDI) Model fit indices: $\chi^2 = 1.477$, df =1, $p = 0.224$, $\chi^2/\text{df} = 1.477$, GFI = 0.998, AGFI = 0.977, CFI = 0.999, RMSEA = 0.038 and SRMR = 0.010 | | | |
| PDI1 | 0.7 | 10.330 | $\alpha = 0.84$ AVE = 0.57 MSV = 0.55 ASV = 0.24 |
| PDI2 | 0.69 | 12.432 | |
| PDI3 | 0.85 | 12.422 | |
| PDI4 | 0.77 | | |
| Process innovation (PCI) Model fit indices: $\chi^2 = 0.004$, df = 1, $p = 0.95$, $\chi^2/\text{df} = 0.004$, GFI = 1.0, AGFI = 1.0, CFI = 1.00, RMSEA = 0.000 and SRMR = 0.000 | | | |
| PCI1 | 0.78 | 15.17 | $\alpha = 0.90$ AVE = 0.70 MSV = 0.59 ASV = 0.26 |
| PCI2 | 0.89 | 16.19 | |
| PCI3 | 0.82 | 15.17 | |
| PCI4 | 0.86 | | |
| Financial perspective (FIP) Model fit indices: $\chi^2 = 1.211$, df =1, $p = 0.271$, $\chi^2/\text{df} = 1.211$, GFI = 0.998, AGFI = 0.981, CFI = 1.000, RMSEA = 0.008 and SRMR = 0.008 | | | |
| FIP1 | 0.88 | 17.59 | $\alpha = 0.91$ AVE = 0.72 MSV = 0.55 ASV = 0.19 |
| FIP2 | 0.93 | 19.08 | |
| FIP3 | 0.72 | 14.11 | |
| FIP4 | 0.86 | | |
| Customer perspective (CUP) Model fit indices: $\chi^2 = 1.53$, df = 1, $p = 0.216$, $\chi^2/\text{df} = 1.530$, GFI = 0.998, AGFI = 0.977, CFI = 0.999, RMSEA = 0.040 and SRMR = 0.007 | | | |
| CUP1 | 0.89 | 9.37 | $\alpha = 0.80$ AVE = 0.51 MSV = 0.48 ASV = 0.12 |
| CUP2 | 0.80 | 9.22 | |
| CUP3 | 0.57 | 9.26 | |
| CUP4 | 0.54 | | |
| Internal process perspective (INP) Model fit indices: $\chi^2 = 1.734$, df =1, $p = 0.188$, $\chi^2/\text{df} = 1.734$, GFI = 0.997, AGFI = 0.973, CFI = 0.998, RMSEA = 0.048 and SRMR = 0.008 | | | |
| INP1 | 0.71 | 12.14 | $\alpha = 0.81$ AVE = 0.52 MSV = 0.50 ASV = 0.27 |
| INP2 | 0.50 | 8.16 | |
| INP3 | 0.75 | 12.46 | |
| INP4 | 0.87 | | |
| Learning perspective (LNP) Model fit indices: $\chi^2 = 0.079$, df = 1, $p = 0.779$, $\chi^2/\text{df} = 0.079$, GFI = 1.0, AGFI = 0.999, CFI = 1.00, RMSEA = 0.001 and SRMR = 0.001 | | | |
| LNP1 | 0.86 | 12.51 | $\alpha = 0.88$ AVE = 0.64 MSV = 0.53 ASV = 0.18 |
| LNP2 | 0.82 | 12.29 | |
| LNP3 | 0.84 | 14.26 | |
| LNP4 | 0.67 | | |

Composite reliability (CR) and average variance extracted (AVE) were calculated following the construction reliability test of [76,78] for scale determination of the final measurement model. As shown in Table 1, overall reliability was between 0.78 and 0.91. For each variable, values greater than 0.7 indicated a good level of confidence. All AVEs were greater than 0.50 (AVE > 0.50), or lower, except DIFs below 0.5. However, Fornell and Larcker [76] stated that AVE could be less than 0.50 if CR was greater than 0.60 and Maximum Shared Variance (MSV) and Average Shared Variance (ASV) could be less than AVE. Thus, the measured convergence accuracy was sufficient for the measurement model and all theoretical structures had acceptable psychological properties.

## 4. Data Analysis

Regarding the descriptive statistics, a quarter of the samples were Tier 1 (57.30%), Tier 2 (31.00%), while the rest were Origianl Equipment Manufacturer (OEM) (11.70%). Their employees ranged at more than 1500 members (45.1%) with most having less than 500 members (21.80%), followed by 501–1000 (20.60%) and 1001–1500 (12.6%). Most had worked for more than 15 years (60.1%), followed by 5–10 years (18.1%), 11–15 years (12.6%), and less than 5 years (9.2%).

The analysis results gave standard deviation (SD) and coefficient of variation (CV) of all indicators at a high level, with the relationship among variables at an average level, as shown in Table 3.

**Table 3.** Descriptive statistics and correlation.

|  | **Mean** | **SD** | **CV** | **DYC** | **CPT** | **INC** | **FRM** |
|---|---|---|---|---|---|---|---|
| DYC | 3.876 | 0.568 | 0.146 | 1 | | | |
| CPT | 3.816 | 0.634 | 0.166 | 0.717 ** | 1 | | |
| INC | 3.727 | 0.728 | 0.195 | 0.689 ** | 0.747 ** | 1 | |
| FRM | 3.830 | 0.591 | 0.154 | 0.663 ** | 0.731 ** | 0.751 ** | 1 |

** $p < 0.001$.

Analysis results of the independent variable showed that DYC had an average of 3.876, whereas the coefficient of variation (CV) was 0.142. CPT and INC as mediators had average values of 3.816 and 3.727, with CV at 0.166 and 0.195, respectively. FRM as the dependent variable had average values of 3.830 with CV at 0.154. When determining the relationships between the four variables in five pairs, the correlation coefficients were between 0.663 and 0.751 at the range of medium to high level. The relationships did not show multicollinearity among latent factors. Tabachnick and Fidell [79] stated that multicollinearity occurs when the correlation coefficient between each pair of variables is more than 0.90. Therefore, results of the relationships among variables conformed to this basic statistical criterion.

Step 1 Analysis of the influence of independent variable and dependent variable followed the concept of Baron and Kenny [80]. They stated that, for analysing the influence of the independent variable (X) which influences the dependent variable (Y) in the path $X \overset{c}{\to} Y$ must be statistically significant. If c is higher than 0.20, considered as abnormally high, then there is evidence of mediation separation. The independent variable was dynamic capabilities and the dependent variable was firm performance. Results showed that dynamic capabilities as the independent variable influenced firm performance as the dependent variable with a 0.69 path coefficient at 0.001 statistical significance, as shown in Figure 2.

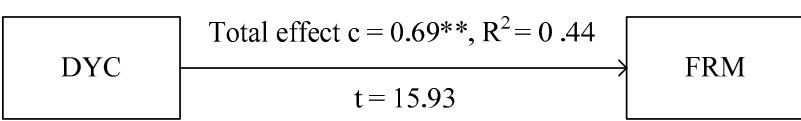

**Figure 2.** Influence of independent variable (DYC) on dependent variable (FRM). Note: ** $p < 0.001$.

When the path coefficient between independent variable and dependent variable was higher than 0.20 following the Baron and Kenny [80] concept, mediator separation occurred between both variables. The mediator was further studied in hypotheses 2 and 3.

Step 2 The influence of the independent variable (X) on the mediator (M) in the path X $\overset{a}{\rightarrow}$ M was tested for statistical significance, with independent variables including competitive advantages and innovation capabilities. Each variable was analysed. Results showed that dynamic capabilities as an independent variable influenced competitive advantages and innovation capabilities as mediators, with path coefficients of 0.82 and 0.89 and statistical significance of 0.001, as shown in Figure 3a,b.

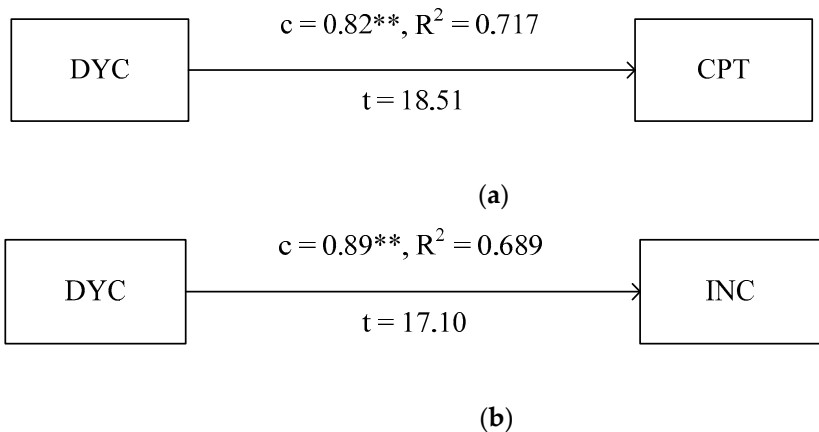

(a)

(b)

**Figure 3.** (**a**) Influence of the independent variable (DYC) on the mediator (CPT). (**b**) Influence of the independent variable (DYC) on the mediator (INC). Note: ** $p < 0.001$.

Step 3 The influence of the mediator (M) on dependent variable (Y) in the path M $\overset{b}{\rightarrow}$ Y was tested for statistical significance with the mediators' competitive advantages and innovation capabilities and dependent variable (Y) as firm performance. Results showed that competitive advantages and innovation capabilities as mediators influenced firm performance as a dependent variable with path coefficients of 0.48 and 0.46 at 0.001 statistical significance, as shown in Figure 4a,b.

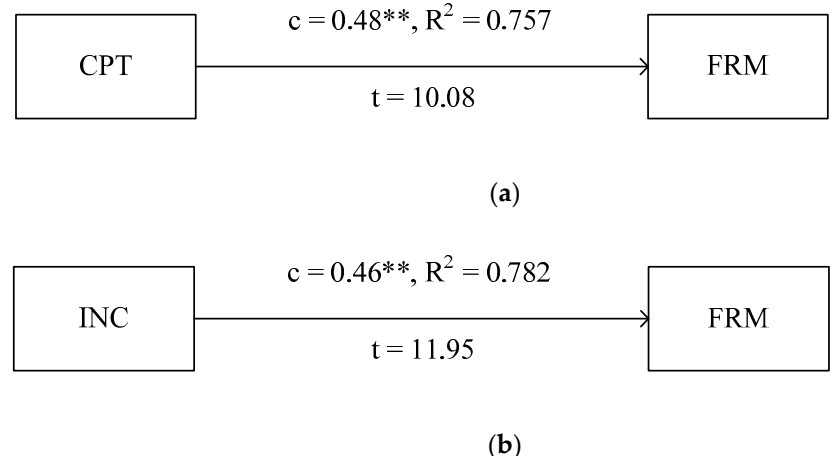

(a)

(b)

**Figure 4.** (**a**) Influence of the mediator (CPT) on the dependent variable (FRM). (**b**) Influence of the mediator (INC) on the dependent variable (FRM). Note: ** $p < 0.001$.

Step 4 When considering the previous test results, test criteria met all conditions for the Baron and Kenny [80] concept. The full model was then analysed using the independent variables including dynamic capabilities, with firm performance as a dependent variable, and then tested for mediators including competitive advantages and innovation capabilities to consider the c' value. This had to be

less than c, as a result of the mediator added between the independent variable and the dependent variable. After analysing the mediator in this model, c was not necessarily significant. In the case where the c' value was still statistically significant, the mediator was called a partial mediator, whereas if c' was not statistically significant, the mediator was called a full mediator, as shown in Figure 5.

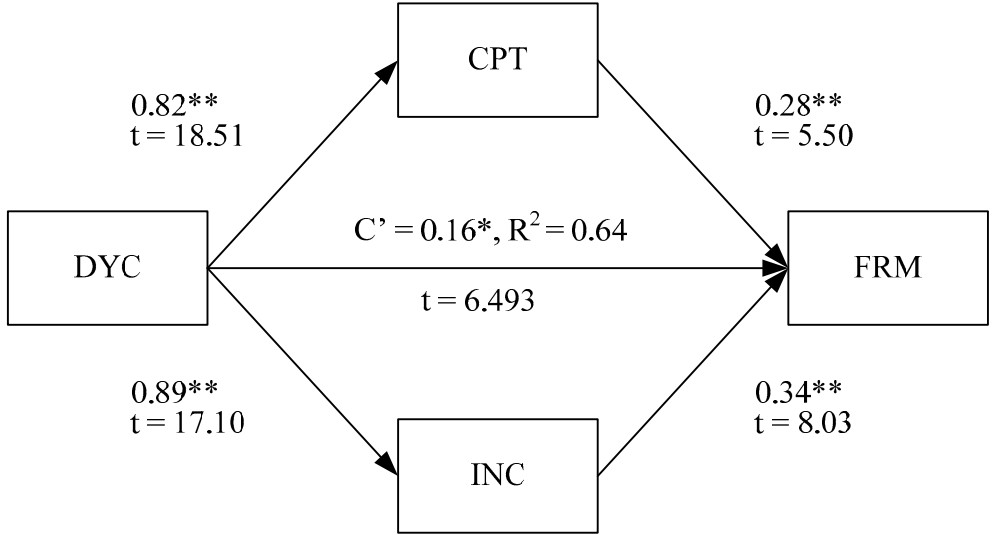

**Figure 5.** Test of the influence of the full model. Note: * $p < 0.05$, ** $p < 0.001$.

Results of step 4 are shown in Figure 5. Dynamic capabilities as an independent variable had a direct influence on firm performance as a dependent variable. The direct influence of the path coefficient decreased from 0.69 to 0.16 at a statistical significance level of 0.001 or 23.19%.

Indirect effect testing was conducted using repeated tests with 5000 exchanges; n was 326 units and sampling with substitutions withstood repeated units. Each data set was used for regression analysis by specifying the independent variable and mediator. Results showed product path coefficient to and from the median and the standard deviation (SD) with a 5000 value and then proceeded to type 1 or type 2 (Table 4).

**Table 4.** Indirect effects through factors of CPT and INC using the bootstrap method.

| Path | Estimate | SE | *t* | *p* | 95% CI | |
|---|---|---|---|---|---|---|
| | | | | | LL | UL |
| DYC→CPT→FRM | 0.3907 | 0.0507 | 5.487 | 0.000 | 0.2965 | 0.4948 |
| DYC→INC→FRM | 0.4085 | 0.0444 | 5.6196 | 0.000 | 0.3299 | 0.5040 |
| DYC→INC/CPT→FRM | 0.5295 | 0.0560 | 3.0163 | 0.0028 | 0.4313 | 0.6503 |

Type 1 calculated the product average of the path coefficient and the next SE mean with significance when $|t| > 2.00$. It had significance at the level of 0.05.

Type 2 represented the ascending product of the path coefficient along the route to and from the mediator (that is, the value of 1000). The range for each of these values was determined from 2.5% to 97.5% to find zero coverage. If covered, zero means that, at a significance level of 5%, the value of the product does not change from zero. Indirect impact analysis was conducted following version 4 of Hayes [81]. Results from the macro version 3.1 of IBM SPSS Statistics 21.0 for Windows are shown in Table 4.

Bootstrapping analysis was conducted using the PROCESS version 3.5 command in the SPSS program. PROCESS can analyse both direct and indirect influences through the transmitted variables. The relationship model 4 used the program to rotate 5000 cycles.

For *t*-values and significance levels, a |*t*| value greater than 2.00 showed that the competitive advantages had an indirect influence on dynamic capabilities and organisational performance at the significance level of 0.05

Indirect influence was estimated using the bootstrap method at the level of 0.05 significance as 95% confidence interval between 0.22 and 0.40 for competitive advantages and 0.14 and 0.33 for innovation capabilities. Results showed that innovation capabilities acted as a mediator between dynamic capabilities and firm performance. This partial influence (partial mediation) confirmed that dynamic capabilities had a direct influence on firm performance by passing on both competitive advantages and innovation capabilities as a partial mediator. Thus, the results supported hypotheses 2 and 3 respectively.

The first research question asked whether CPT and INC were mediators between DYC and FRM. Analysis results suggested that CPT and INC were real mediators (partial mediation) between DYC and FRM.

The second research question concerned the relationship between DYC and FRM. Results indicated that DYC had both a direct and indirect influence on FRM.

## 5. Discussion

Results indicated that DYC had a direct effect on FRM. This can be explained because, with more knowledge concerning management change, organisations can improve and achieve FRM targets. Teece and Leih [8] and Fainshmidt and Pezeshkan [82] emphasised the positive relationship between dynamic capabilities and firm performance. However, Fainshmidt and Pezeshkan [82] suggested that the relationship between dynamic capabilities and firm performance depended on contextual factors. Girod and Whittington [83] discussed whether dynamic capabilities facilitated competitive advantages and higher levels of performance in a stable or dynamic work environment.

DYC direct-affected FRM with mediators as CPT and INC. Similarly, Pisano [84] found that, under a changing environment, dynamic capabilities were required to choose the right resources to create competitive advantages leading to good organisational performance. In the same vein, Navarro-García and Moreno [50], Breznik and Lahovnik [85] determined that dynamic capabilities influenced organisational performance by passing on competitive advantages. An organisation with dynamic capabilities has the potential to maintain its competitive advantages, leading to future success.

Huse [30] and Popa, and Soto-Acosta [31] found that the relationship between dynamic capabilities and innovation capabilities stimulated and increased innovation activities, resulting in successful performance. The dynamic of customer needs allows organisations to strive to submit ideas and present innovative products or services that can create greater value [32].

## 6. Conclusions

DYC may not be the only factor affecting FRM. Previous studies and behavioural theories indicate that CPT and INC as mediators also affect FRM. The study emphasised on only one variable to another and did not investigate two mediate variables simultaneously. This empirical study suggests that organisations need to focus on their ability to cope with change. FRM will be high if supported by DYC and FRM will be enhanced if it is supported by DYC, CPT, and INC. FRM will also improve if DYC are enhanced to promote CPT and INC.

The dynamic capabilities perspective is a reaction to a defect of the resource-based view in the new economic environment of innovative technology. Dynamic capabilities are defined by three distinct dimensions: sensing capability, seizing capability, and transforming capability. Theoretically, the extant literature suggests that dynamic capabilities do not adapt enough to environment change. There needs to be another factor to support the ability to improve more than usual.

Many definitions of dynamic capabilities point to the importance of innovation capability as well as product and process. However, the concept of dynamic talent is broad enough to provide for comparative support as a leader in differentiation and cost.

The objective of this article was to resolve this controversy by proposing and testing a mediating mechanism between dynamic capacity and firm performance of an organization. The results show that innovation capability and competitive advantage act as a mediating mechanism between a dynamic capability and firm performance. Both mediators are widely considered to be key mechanisms for organization to adapt and define the environment in which they operate. Our findings suggest that, despite subtle differences in the dimensions of dynamic capabilities, innovation capability, and competitive advantage, it also acts as a mediator for improving the efficiency of the organization.

The growth of electric vehicles is affecting the domestic automotive industry, with continuous introduction of new technology. Organisations need to adjust quickly to keep pace with this rapid growth and change. The resource-based perspective theory considers fundamental ability that is used on a regular basis. This may not be sufficient to manage changes and develop new capabilities. Organisations must pay attention to dynamic capabilities that are superior to basic abilities and monitor the changing environment; the organization needs to be able to manage the change under the changing environment if the automotive industry wants to survive under these environmental fluctuations. Just developing the basic abilities may not be enough for the changing environment. Organisations should focus on the theory of dynamic capabilities along with developing competitive advantages in cost leadership and differentiating aspect and innovation capabilities in product innovation and process innovation aspect as a strategy to attain the desired results.

### 6.1. Theoretical Implications

This research presents a model showing the interaction between dynamic capabilities and firm performance that directly affects and influences the transmission of competitive advantages and innovation capabilities to enhance knowledge. Results can be used as a guideline for further studies to improve and develop future theoretical concepts adapted for other situations and industries.

### 6.2. Practical Implications

The global automotive industry trends are transforming from internal combustion engines (ICE) technology; some countries have had expertise and capacity to manufacture parts over the past 100 years. It has been replaced by electric vehicles and self-driving vehicles (autonomous vehicle), seen as disruption technology in the future of the automotive industry. The change is having an impact on the traditional automotive supply chain, including automakers and auto parts manufacturers. The production potential of the traditional automobile industry is not ready to support the production of electric vehicle parts. Regardless of the labour market, a lack of specialized skills in electric vehicles, resources for battery and electronic assembly, research and development of related innovative technologies and investment in industrial development require large investment in new industries. This includes government policies on the demand chain, such as providing benefits to consumers who decide to buy electric vehicles in the country. The government's role should be changing from regulation control toward facilitation. Government has become proactively engaged with all stakeholders, from technology transfer to knowledge co-creation, and set up new automotive regulation to cover the dynamic situation.

Hence, governments need to focus on increasing their capacity to cope with disruptive innovation by promoting dynamic capabilities such as Sensing, Seizing, and Transforming with all sub-economies such as SME, start-up, big business, and social open innovation to cause a balance dynamic innovation effect across the economic cycle. This includes promoting the creation of innovations capability that responds to product innovation to customers' needs and process innovation to create manufacturing processes that were consistent with modern products and cost-effective, and promoting the enhancement of Competitive Advantages that respond to cost leadership and differentiate that enable organizations to compete in the automotive industry and other industries, such as the aircraft parts industry and the medical industry. This approach will meet the industry needs and changes to increase the

competitiveness of the organization, both inside and outside the country. This approach will meet the industry needs and changes to increase competitiveness in the country.

Results can be used as guidelines for companies in the Thai automotive industry to assess their dynamic capabilities and increase their ability to adapt to the electric vehicle market. Companies can use these research results as a foundation to first understand and then plan and determine ways to develop in-house dynamic ability. Continuous adaptation to changing environmental and technological conditions will ensure efficiency and success.

*6.3. Research Limitations*

This research has some limitations. A cross-sectional study investigates a specific period of time. Therefore, longitudinal studies are recommended for future research to verify the reliability of the results. Competitive advantages and innovation capabilities were used as mediators in the context of the Thai automotive industry that is different from other industries in disparate countries. The educational context should be extended to cover varying industries, societies, and cultures to increase the strength of the model. Moreover, only two mediators were confirmed, so prediction results were not high. Future research should include other factors gained from interviews with central administrators or government officials.

**Author Contributions:** Conceptualization, A.R., P.S. and K.N.-N.; methodology, A.R., P.S. and K.N.-N.; software, A.R. and K.N.-N.; validation, A.R., P.S. and K.N.-N.; formal analysis, A.R., P.S. and K.N.-N.; investigation, A.R. and K.N.-N.; resources, A.R., P.S. and K.N.-N.; data curation, A.R. and K.N.-N.; writing—original draft preparation, A.R.; writing—review and editing, A.R., P.S. and K.N.-N.; visualization, A.R., P.S. and K.N.-N.; supervision, A.R., P.S. and K.N.-N.; project administration, A.R. All authors have read and agreed to the published version of the manuscript.

**Funding:** This research received no external funding.

**Conflicts of Interest:** The authors declare no conflict of interest.

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
