# Peer review of "Development of Dynamic Capabilities for Automotive Industry Performance under Disruptive Innovation"

_2199-8531, doi:10.3390/joitmc6040097_

Round 1

Reviewer 1 Report

The topic of the paper is innovative and the processing of the paper is in accordance with contemporary state of knowledge in scientific community. However, the literature sources are mainly old, so the literature review does not form an adequate basis for the upcoming research. Thus, it is vital to extend used sources and focus on latest journal publications indexed in WoS resp. SCOPUS in quartile Q1 resp. Q2.

The title of the paper is well formulated and it covers the content of the paper. The abstract contains main required components (theme, procedure, main findings). The paper is written in accordance with traditional IMRD (introduction, methodology, results, discussion) structure of scientific papers. The methods are stated clearly and it is not possible to consider them as sufficient. The level of author’s knowledge is satisfying. It is obvious that authors are well oriented in the topic and that they use appropriate terms. Overall, I was interested to read this manuscript and I believe that it is worthy of a publication in this journal after minor corrections.

I wish the author(s) the best of luck with their future research.

Author Response

The topic of the paper is innovative and the processing of the paper is in accordance with contemporary state of knowledge in scientific community. However, the literature sources are mainly old, so the literature review does not form an adequate basis for the upcoming research. Thus, it is vital to extend used sources and focus on latest journal publications indexed in WoS resp. SCOPUS in quartile Q1 resp. Q2.

Action/Answer: Thank you so much for your kind suggestion. We updated the new literature sources on pages: 2,3,5 as follow

  1. Hermawati, A., The implementation of dynamic capabilities for SMEs in creating innovation.Journal of Workplace Learning, 2020.
  2. Garbellano, S. and M.d.R. Da Veiga, Dynamic capabilities in Italian leading SMEs adopting industry 4.0.Measuring Business Excellence, 2019.
  3. Strøm-Andersen, N., Incumbents in the Transition Towards the Bioeconomy: The Role of Dynamic Capabilities and Innovation Strategies.Sustainability, 2019. 11(18): p. 5044.
  4. Arranz, N., et al., Innovation as a driver of eco‐innovation in the firm: An approach from the dynamic capabilities theory.Business Strategy and the Environment, 2020. 29(3): p. 1494-1503

The title of the paper is well formulated and it covers the content of the paper. The abstract contains main required components (theme, procedure, main findings). The paper is written in accordance with traditional IMRD (introduction, methodology, results, discussion) structure of scientific papers. The methods are stated clearly and it is not possible to consider them as sufficient. The level of author’s knowledge is satisfying. It is obvious that authors are well oriented in the topic and that they use appropriate terms. Overall, I was interested to read this manuscript and I believe that it is worthy of a publication in this journal after minor corrections.

I wish the author(s) the best of luck with their future research.

Action/Answer: I am very pleased to have your kind suggestion to improve my manuscript.

Reviewer 2 Report

The authors of the paper Development of dynamic capabilities for automotive industry performance under disruptive innovation present a relevant topic especially in the current conditions of automotive industry changes in which the competitive advantage is related to the operational and organizational capacity of companies.

Bibliographic sources, citations, and literature are adequately mentioned in the paper, the authors of the research approached the explanations starting from the definitions of each word (such as "capabilities" and "dynamic", through the works of Jantunen, A. at all, 2012 and McKelvie, A. and P. Davidsson, 2009).

The research methodology is appropriate, the authors use the online survey as a research tool, the sample size being 326 companies. Moreover, for the data evaluation and analysis process, variables were identified to test the proposed relationships, and for statistical analysis, SPSS 26, AMOS 21, and PROCESS macro 3.6 were used.

The research results are presented by the research authors both theoretically and especially pragmatically, highlighting the competitive advantages and innovation capabilities partially mediated by the dynamic capabilities and performance of companies. Moreover, the authors emphasize that competitive advantages and innovation capabilities act as mediators to improve the dynamic capabilities and increase the innovative performance of firms, which is justified by the authors as a contribution to the literature. This aspect is pragmatically argued by the authors of the research, but we appreciate that this personal contribution should also be argued from a scientific point of view. Therefore, we suggest the authors complete in the results chapter also the personal scientific contribution (either conceptually or from the point of view of the scientific model resulting from the research carried out).

The conclusions are properly presented, the authors capture the theoretical and practical implications, as well as the limitations of the study and future research, but as we mentioned in the Results chapter, we suggest research authors complete the conclusions with the results Specialty literature.

We congratulate the research team, we suggest the revision of the chapters of results and conclusions, and after the revision, we propose for acceptance of the paper.

Author Response

The authors of the paper Development of dynamic capabilities for automotive industry performance under disruptive innovation present a relevant topic especially in the current conditions of automotive industry changes in which the competitive advantage is related to the operational and organizational capacity of companies.

Bibliographic sources, citations, and literature are adequately mentioned in the paper, the authors of the research approached the explanations starting from the definitions of each word (such as "capabilities" and "dynamic", through the works of Jantunen, A. at all, 2012 and McKelvie, A. and P. Davidsson, 2009).

The research methodology is appropriate, the authors use the online survey as a research tool, the sample size being 326 companies. Moreover, for the data evaluation and analysis process, variables were identified to test the proposed relationships, and for statistical analysis, SPSS 26, AMOS 21, and PROCESS macro 3.6 were used.

The research results are presented by the research authors both theoretically and especially pragmatically, highlighting the competitive advantages and innovation capabilities partially mediated by the dynamic capabilities and performance of companies. Moreover, the authors emphasize that competitive advantages and innovation capabilities act as mediators to improve the dynamic capabilities and increase the innovative performance of firms, which is justified by the authors as a contribution to the literature. This aspect is pragmatically argued by the authors of the research, but we appreciate that this personal contribution should also be argued from a scientific point of view. Therefore, we suggest the authors complete in the results chapter also the personal scientific contribution (either conceptually or from the point of view of the scientific model resulting from the research carried out).

Action/Answer: I am very pleased to have your kind suggestion to improve my manuscript. This manuscript focuses on the Quantitative research using data collection by questionnaire distribution methods. For my next research, I will focus on Qualitative research by in depth interviewing executives who are relevant in the automotive industry.

The conclusions are properly presented, the authors capture the theoretical and practical implications, as well as the limitations of the study and future research, but as we mentioned in the Results chapter, we suggest research authors complete the conclusions with the results Specialty literature.

We congratulate the research team, we suggest the revision of the chapters of results and conclusions, and after the revision, we propose for acceptance of the paper.

Action/Answer: Thank you so much for your kind suggestion. Revision has been revised on page16-17 as follow

The dynamic capabilities perspective is a reaction to a defect of resource-based view in new economic environment of innovative technology. Dynamic capabilities are defined by three distinct dimensions: sensing capability, seizing capability and transforming capability. Theoretically, the extant literature suggests that dynamic capabilities not enough adapt to environment change. There needs to be another factor to support the ability to improve more than usual.

Many definitions of dynamic capabilities point to the importance of innovation capability as well as product and process. However, the concept of dynamic talent is broad enough to provide for comparative support as a leader in differentiation and cost.

The objective of this article is to resolve this controversy by proposing and testing a mediating mechanism between dynamic capacity and firm performance of organization. The results show that innovation capability and competitive advantage act as a mediating mechanism between a dynamic capability and firm performance. Both mediators are widely considered to be key mechanisms for organization to adapt and define the environment in which they operate.  Our findings suggest that despite subtle differences in the dimensions of dynamic capabilities, innovation capability and competitive advantage, but it also acts as mediator for improving the efficiency of the organization.

Reviewer 3 Report

However, this is managed as a new submission; let me refer back to the process of the previous process. The authors made an interesting analysis using a very popular method. First, I asked for doing major amendments since the conclusion was weak. The improved version of the paper fulfilled my requirements, and the authors made changes asked by other reviewers. The newly submitted paper confirms my opinion about accepting the paper for publication. I can see further development by the instructions.

Author Response

However, this is managed as a new submission; let me refer back to the process of the previous process. The authors made an interesting analysis using a very popular method. First, I asked for doing major amendments since the conclusion was weak. The improved version of the paper fulfilled my requirements, and the authors made changes asked by other reviewers. The newly submitted paper confirms my opinion about accepting the paper for publication. I can see further development by the instructions.

Action/Answer: I am very pleased to have your kind suggestion to improve my manuscript.

Reviewer 4 Report

The motivation of the manuscript entitled: “Development of dynamic capabilities for automotive industry performance under disruptive innovation” deals with a very important topic. I appreciate the aims of this work; it is quite interesting and informative to most readers of this field. Novelty and contribution are unquestionable. In general, it is well grounded in the body of knowledge in the area. The background of the manuscript is clearly explained. The theory is presented in an acceptable way even for readers not familiar with these issues.

Thus, in my opinion the manuscript is recommendable for publication.

Author Response

The motivation of the manuscript entitled: “Development of dynamic capabilities for automotive industry performance under disruptive innovation” deals with a very important topic. I appreciate the aims of this work; it is quite interesting and informative to most readers of this field. Novelty and contribution are unquestionable. In general, it is well grounded in the body of knowledge in the area. The background of the manuscript is clearly explained. The theory is presented in an acceptable way even for readers not familiar with these issues.

Thus, in my opinion the manuscript is recommendable for publication.

Action/Answer: I am very pleased to have your kind suggestion to improve my manuscript.

This manuscript is a resubmission of an earlier submission. The following is a list of the peer review reports and author responses from that submission.

Round 1

Reviewer 1 Report

The title of the paper is well formulated and it covers the content of the paper. The contribution of the paper is significant in scope of contemporary state of knowledge. The abstract contains main required components (theme, procedure, main findings) and the goal of the paper is stated clearly. Methodological part of the paper is rather poor and description of deeper statistical background as well as relevant statistical formulas are missing at all. Discussion does contain constructive comparison of results of own research with contemporary state of knowledge formulated so far. Theoretical implications, practical implications and research limitations are incorporated correctly. However, possible future perspectives of the research should be also included. Authors of the paper don´t use recent literature. It would be highly appreciated to enrich the literature and to include also references from journals indexed in reputable databases (WoS resp. Scopus) in last two years. However, the level of the author’s knowledge is satisfying. It is obvious that authors are well oriented in the topic and that they use appropriate terms. The overall level of language is appropriate. Quotations respect APA style. Writing style is clear and understandable. The paper would be suitable to be published in the journal after minor revisions provided in accordance with this review.

Author Response

Thank you so much for your kind suggestion.

Point 1:Methodological part of the paper is rather poor and description of deeper statistical background as well as relevant 

Response 1:

For methodological part revision has been revised on page 11-12 as below.

The study population was 833 companies from the Thai automotive industry (data as of July 29, 2019). The sample size of 220 companies was determined using the criteria of Hair, Black [46]. They specified that the ratio between the number of samples and the number of parameters should be 25:1 to be sufficient for confirmatory factor analysis (CFA). Eleven observable variables were identified. Therefore, the sample group must be at least 275 companies. For convenience of evaluation and data analysis, the sample size was finalised at 326 companies. The sample size of 326 companies selected by use of Stratified random sampling. The calculation resulted in 275 sample units. Convenience sampling was followed by email as effective way for supporting variety and qualifications of respondents. An online survey has more advantages than post or paper surveys because of the capability for quick replies without the limitation of geography. Therefore, for the first round, 400 survey emails were sent to all samples and 281 replies were received within four weeks. Survey emails were then sent to another 100 samples and 45 replies were sent back within four weeks to give a total of 326 respondents.

Point 2:Authors of the paper don´t use recent literature.

Response 2:

recent literature revision has been revised on page 7, 10 and 14. as below.

18.Na-nan, K., K. Chaiprasit, and P. Pukkeeree, A validation of the performance management scale. International Journal of Quality & Reliability Management, 2018. 35(6): p. 1253-1267.

19.Na-Nan, K., Organizational behavior. 2020, Bangkok: Triple Education Co., Ltd.

45.Na-Nan, K. and E. Sanamthong, Self-efficacy and employee job performance: Mediating effects of perceived workplace support, motivation to transfer and transfer of training. International Journal of Quality & Reliability Management, 2019. 37(1): p. 1-17.

58.Na-Nan, K., Organizational behavior scale developement. 2020, Bangkok: Triple Education Co., Ltd.

Reviewer 2 Report

The paper offers an interesting insight into the impact of innovation on financial performance. Never the less, justification for selecting Thai automotive industry needs to be provided.How is this article  different from: Chamsuk, W., Fongsuwan, W. and Takala, J., 2017. The effects of R&D and innovation capabilities on the Thai automotive industry part’s competitive advantage: a sem approach? Or Sueptaetrakun, W., 2018. Determinants of collaboration and innovation towards organization performance within Thailand's automotive parts industry. Journal of Business and Retail Management Research13(2) for that matter is not very clear.

Should dynamic and innovation capabilities be two separate factors? A clearer indication of Discriminant Validity

·         MSV < AVE ·         ASV < AVE needed to be provided. Theoretical and practical implications are severely under developed. It is stated on page 16: " Government agencies involved in industrial supervision and promotion can use this data to establish policy direction and develop, promote and support the Thai automotive industry. " Adequate background and research problem have not been provided. Thus, such practical implications appear vague. There is a brief mention of innovation in the Conclusions section. It is not adequate as dynamic and disruptive innovations needed to be explained from the context of methodology adopted for the  purpose of this research.

Author Response

Thank you so much for your kind suggestion.

Point 1:How is this article  different from: Chamsuk and Sueptaetrakun. 

Response 1: After compare 2 papers we’re focusing on automotive industry and using Innovation in mediator analysis but we’re difference dependent variably and independence variably and my paper is focusing to Results showed that innovation capabilities acted as a mediator between dynamic capabilities and firm performance. This partial influence (partial mediation) confirmed that dynamic capabilities had a direct influence on firm performance by passing on both competitive advantages and innovation capabilities as a partial mediator.

Point 2: Additional MSV and ASV 

Response 2:Additional MSV and ASV revision has been revised on page 15-20 as below.

Composite reliability (CR) and average extraction difference (AVE) were calculated following the construction reliability test of [53] for scale determination of the final measurement model. As shown in Table 1, overall reliability was between 0.78 and 0.91. For each variable, values ​​greater than 0.7 indicated a good level of confidence. All AVEs were greater than 0.50 (AVE> 0.50), or lower, except DIFs below 0.5. However, Fornell and Larcker [53]  stated that AVE could be less than 0.50  if CR was greater than 0.60 and Maximum Shared Variance (MSV) and Average Shared Variance (ASV) could be less than AVE. Thus, the measured convergence accuracy was sufficient for the measurement model and all theoretical structures had acceptable psychological properties.

Point 3: Adequate background and research problem have not been provided

Response 3:

Adequate background and research problem revision has been revised on page 1-2 as below.

The automotive industry is a major industry that has driven the economy of Thailand, which has been supported by the government since the Second National Economic and Social Development Plan. Make Thailand become one of the world's major automotive car maker and automotive parts manufacturers. Due to the rapidly changing air pollution and natural environment, worldwide are looking for vehicles that are more energy-efficient and environmentally friendly. For this reason, the demand for vehicle in the world market has changed from the internal combustion engine vehicles, in which Thailand has expertise in both assembly and production of the change to new demand for electric vehicle in Thailand is still in the period of development plan only. Electric vehicles are seen as a disruptive technology that will make a difference and affect the automotive industry worldwide. The supply chain of Thai automotive industry has to face modern technology that comes with electric vehicles. The Resources Base View and Operation Capability that the organization has under normal conditions may not be sufficient under the environmental turbulence situation. Dynamic capability is therefore an important role in managing the organization's strategy to be suitable for the competitive and highly dynamic situations. This article would like to presents guidelines for development of dynamic capability in Thai automotive industry, in terms of sensing, seizing and transforming to be a guideline for preparing the adaptation of the Thai automotive industry to keep up with the changes that will occur in the future.

Reviewer 3 Report

The paper investigates a relevant topic that may be of broader interest to the readers. The paper is well structured; the scope and the deepness of the analysis is appropriate. However, I have to ask some revision marked as a major, but these do not require essential changes. Revision is required according to the following issues:

  • The format of the abstract is not usual in this journal. I find the highlight of the topics in the abstract unmercenary, rewrite in in more simple structure.
  • H1 hypothesis need a better explanation. It can be understood in several ways. The wording does not match the good quality of the analysis.
  • Figure 1 is broken. Reformat it, and I suggest pasting it as a picture.
  • Discussion and especially conclusions are weak. It is usual in similar papers that there is a great model with a significant result than a short closing. Use the model as a tool for making your statements and use it just to support what you have to say. Add more details in the conclusion section.

Author Response

Thank you so much for your kind suggestion

Point 1: Discussion and especially conclusions are weak. It is usual in similar papers that there is a great model with a significant result than a short closing. 

Response 1: Practical implications and conclusions revision has been revised on page 13-14 as below.

  1. Discussion

Results indicated that DYC had a direct effect on FRM. This can be explained because with more knowledge concerning management change, organisations can improve and achieve FRM targets. Teece and Leih [7], Fainshmidt, Pezeshkan [62]  emphasised the positive relationship between dynamic capabilities and firm performance. However, Fainshmidt, Pezeshkan [62] suggested that the relationship between dynamic capabilities and firm performance depended on contextual factors. Girod and Whittington [63], discussed whether dynamic capabilities facilitated competitive advantages and higher levels of performance in a stable or dynamic work environment.

       DYC direct affected FRM with mediators as CPT and INC. Similarly, Pisano [64] found that under a changing environment, dynamic capabilities were required to choose the right resources to create competitive advantages leading to good organisational performance. In the same vein, Navarro-García, Moreno [65], Breznik and Lahovnik [66] determined that dynamic capabilities influenced organisational performance by passing on competitive advantages. An organisation with dynamic capabilities has the potential to maintain its competitive advantages, leading to future success.

       Huse [35] and Popa, Soto-Acosta [36] found that the relationship between dynamic capabilities and innovation capabilities stimulated and increased innovation activities, resulting in successful performance. The dynamic of customer needs allows organisations to strive to submit ideas and present innovative products or services that can create greater value [34]

6.1. Theoretical implications

This research presents a model showing the interaction between dynamic capabilities and firm performance that directly affects and influences the transmission of competitive advantages and innovation capabilities to enhance knowledge. Results can be used as a guideline for further studies to improve and develop future theoretical concepts adapted for other situations and industries.

6.2. Practical implications

Government agencies involved in industrial supervision and promotion can use this data to establish policy direction and develop, promote and support the Thai automotive industry. The government needs to focus on the development of the Dynamic capability of the automotive industry in order to be aware of the situation and be able to adjust the organization to keep up with the changing of the electric vehicle industry that will replace the internal combustion engine including the growth of the automotive industry as a way to manage energy demand in the future as well. This approach will meet the industry needs and changes to increase competitiveness in the country.

Results can be used as guidelines for companies in the Thai automotive industry to assess their dynamic capabilities and increase their ability to adapt to the electric vehicle market. Companies can use these research results as a foundation to first understand and then plan and determine ways to develop in-house dynamic ability. Continuous adaptation to changing environmental and technological conditions will ensure efficiency and success.

6.3. Research limitations

This research has some limitations. A cross-sectional study investigates a specific period of time. Therefore, longitudinal studies are recommended for future research to verify the reliability of the results. Competitive advantages and innovation capabilities were used as mediators in the context of the Thai automotive industry that is different from other industries in disparate countries. The educational context should be extended to cover varying industries, societies and cultures to increase the strength of the model. Moreover, only two mediators were confirmed, so prediction results were not high. Future research should include other factors gained from interviews with central administrators or government officials.

  1. Conclusions

DYC may not be the only factor affecting FRM. Previous studies and behavioural theories indicate that CPT and INC as mediators also affect FRM. This study emphasised on only one variable to another and did not investigate two mediate variables simultaneously. This empirical study suggests that organisations need to focus on their ability to cope with change. FRM will be high if supported by DYC and FRM will be enhanced if it is supported by DYC, CPT and INC. FRM will also improve if DYC are enhanced to promote CPT and INC.

The growth of electric vehicles is affecting the domestic automotive industry, with continuous introduction of new technology. Organisations need to adjust quickly to keep pace with this rapid growth and change. The resource based perspective theory considers fundamental ability that is used on a regular basis. This may not be sufficient to manage changes and develop new capabilities. Organisations must pay attention to dynamic capabilities that are superior to basic abilities and monitor the changing environment. Which the organization needs to be able to manage the change under the changing environment. If the automotive industry wants to survive under these environmental fluctuations, by just developing the basic abilities may not be enough for the changing environment. Organisations should focus on the theory of dynamic capabilities along with developing competitive advantages in cost leadership and  differentiate aspect and innovation capabilities in product innovation and process innovation  aspect as a strategy to attain the desired results.

Reviewer 4 Report

The authors of the paper Development of dynamic capabilities for automotive industry performance under disruptive innovation presents a relevant topic on "exploring the influences of competitive advantages and innovation capabilities as mediators", especially in the context of improving the innovative performance of companies.

Concepts, bibliographic sources, and citations are appropriately mentioned in the paper. For example, the authors refer to the fact that dynamic capabilities can be used to summarize and create conceptual variables (Drnevich & Kriauciunas, 2011; Jantunen et al., 2012; Teece et al., 1997).

The research methodology was based on the research tool "questionnaire" for testing the relationships proposed in the research, as well as for statistical analysis. Moreover, the authors of the research, the measurement of convergence accuracy uses the measurement model based on indices, so that all theoretical structures had confirmed an acceptable psychological level of properties.

The research results are properly presented, the authors of the research reflecting the fact that dynamic capabilities as an independent variable had a direct influence on the firm performance as a dependent variable. Moreover, the results presented by the research authors show the product path coefficient to and from the median and standard deviation (SE), and the bootstrapping analysis was performed using the PROCESS version 3.5 command from the SPSS program. THE PROCESS can analyze direct and indirect influences through transmitted variables. The results mentioned by the research authors revealed that competitive advantages and innovation capabilities were partially mediated by dynamic capabilities and firm performance. However, the authors of the research consider that they should mention very clearly the personal scientific contribution to the literature as a result of the results obtained.

The conclusions are presented by the authors of the research, but we suggest to the authors of the research to present both personal scientific contributions to the literature as mentioned and in the results chapter), as well as the fact that the authors of the research will continue research in the field.

We congratulate the research team for the study carried out, and after reviewing with additional additions to the resulting chapters and conclusions, we propose for acceptance and publication of the paper.

Author Response

Thank you so much for your kind suggestion

Point 1: the authors of the research consider that they should mention very clearly the personal scientific contribution to the literature as a result of the results obtained.

Response 1:Due to this research is quantitative research. the part of the personal scientific will be proceed in further Qualitative research.

Round 2

Reviewer 2 Report

Thank you for undertaking the revisions. Never the less, the revisions undertaken are not adequate and are quite superficial. The mediating role is not very clear. Abstract represents a very confusing depiction of variables and their impacts. 

Contribution to literature is not adequate. Although justification for Thai context has been added, it is not enough.

Implications regarding dynamic innovation and the concept itself  is not explained very well.

Author Response

Thank you so much for your kind suggestion.

Point 1: The mediating role is not very clear

Response 1: For mediating role revision has been revised on page 3-8.

Point 2: Abstract represents a very confusing depiction of variables and their impacts

Response 2: For Abstract revision has been revised on page 1 as below

Disruptive innovation creating dramatic change for the industry around the world. Resource Base View (RBV) theory and Operational capability theory are the basic capabilities of an organization under normal changing environmental. This creates a competitive advantage and organizational success in a relatively short period of time, in which the dynamic environment is not sufficient to cope with this change. Dynamic capability is a concept for managing change under this dynamic environment. From the past research supports a direct positive relationship between dynamic capability and firm performance but it did not focus on the mediator variables. This research emphasizes the influences of competitive advantages and innovation capabilities as mediators of dynamic capabilities and firm performance were investigated. A cross-sectional design study was utilised and questionnaires were submitted to 326 firms to test the proposed relationships. SPSS 26, AMOS 21 and PROCESS macro 3.6 were used for statistical analysis. Results revealed that competitive advantages and innovation capabilities were partially mediated by dynamic capabilities and firm performance. Findings contribute to the literature on empowering leadership and innovative firm performance by highlighting that competitive advantages and innovation capabilities act as mediators to improve dynamic capabilities and enhance innovative firm performance.

Point 3: Implications regarding dynamic innovation and the concept itself  is not explained very well.

Response 3: For Practical implications revision has been revised on page 19-20 as below.

The global automotive industry is in transition between the two paradigms from Internal Combustion Engines (ICE) technology [73], there are some countries have expertise and capacity to manufacture parts over the past 100 years. It has been replaced by electric vehicle and self-driving vehicle (autonomous vehicle) seen as disruption technology in the future of automotive industry. The change is having an impact on the traditional automotive supply chain, including automakers and auto parts manufacturers. The production potential of the traditional automobile industry is not ready to support the production of electric vehicle parts. Regardless of labor market, lack of specialized skills in electric vehicles, resource for battery and electronic assembly, Research and Development (R&D) of related innovative technologies and investment in industrial development that require large investment in new industries. This includes government policies on the demand chain, such as providing benefits to consumers who decide to buy electric vehicles in the country. Hence, governments should be focusing on increase their capacity to cope with disruption innovation change by promoting dynamic capabilities such as awareness, planning and organizational improvement to the disruption technology. Including promoting the creation of new innovations that respond to the needs of the customers and create them according to the situation and promoting the enhancement of the competitiveness of the organization both domestic and export of the country. This approach will meet the industry needs and changes to increase the competitiveness of the organization both inside and outside the country. This approach will meet the industry needs and changes to increase competitiveness in the country.

                Results can be used as guidelines for companies in the Thai automotive industry to assess their dynamic capabilities and increase their ability to adapt to the electric vehicle market. Companies can use these research results as a foundation to first understand and then plan and determine ways to develop in-house dynamic ability. Continuous adaptation to changing environmental and technological conditions will ensure efficiency and success

Reviewer 3 Report

The authors followed the instructions of the first-round reviews, the conclusions are reconsidered. The results are clear and understandable. I have no further requests.

According to the method, I have a note. The path analysis is a risky method in the sense that numerical results are very sensitive to the sample and the analysis content selection. Although it is very popular in various fields, the interpretation can be limited. Papers are widely criticized, often from me as well if the conclusions are not correctly formed. Along with this, the method is ready to point out interdependencies and impact chains. Considering these characteristics, I find the research and the paper acceptable for publishing.

Author Response

Thank you so much for your kind suggestion.

We've improve our conclusion and implication on page 19-20

Round 3

Reviewer 2 Report

I am sorry but there is not enough differentiation demonstrated between dynamic capabilities and innovation capabilities to warrant innovation capabilities acting as a mediator variable in the model.

The conceptual model does not make sense in its current form.

In the conclusion section: " ...could be focusing on increase their capacity to cope with disruption innovation change by promoting dynamic capabilities such as awareness, planning and organizational improvement to the disruption technology." Once again, this is a superficial addition and variables definitions and measurements are not adequate.

It would have made more sense to combine Sensing (SEN)
Seizing (SEI) Transforming (TRN) dynamic capabilities with the two types of innovation ( product and process) innovations.

The model is potentially flawed in its current form.